# Emotional Dysfunction and Interoceptive Challenges in Adults with Autism Spectrum Disorders

**DOI:** 10.3390/bs13040312

**Published:** 2023-04-05

**Authors:** Saray Bonete, Clara Molinero, Daniela Ruisanchez

**Affiliations:** Department of Psychology, Universidad Francisco de Vitoria, 28223 Pozuelo de Alarcón, Spain

**Keywords:** interoception, alexithymia, emotional regulation, autism spectrum disorder

## Abstract

People with autism spectrum disorder (ASD) frequently show impaired sensory processing in different senses, including the interoceptive system. Recent findings suggest that interoception is a fundamental component of emotional experience and that impaired interoception is associated with alexithymia. This study aims to explore the association and interrelation between interoceptive confusion, alexithymia, and the capacity for emotional regulation among a sample of 33 adults with ASD compared to a control group of 35 adults with neurotypical development and its mutual impact. The participants answered a series of questionnaires addressing these three variables. The results showed (1) significant differences between the groups in all dimensions, with dysfunctional emotional regulation, impaired interoception, and alexithymia in the ASD group, (2) significant correlations between interoceptive confusion, emotional clarity, and alexithymia in the ASD group but only positive correlations between interoceptive confusion and alexithymia in the CG, and (3) that emotional clarity, alexithymia, and autism explain 61% of the variance in interoceptive confusion. These results are in line with previous studies and suggest that training interoceptive ability may enhance emotional clarity and reduce alexithymia among those diagnosed with ASD, with significant implications in the planning of treatment.

## 1. Introduction

Autism spectrum disorder (ASD) is a neurodevelopmental disorder characterized by difficulties in social interaction and communication, as well as restricted or repetitive behaviors or interests. Since the publication of the DSM-5 [1], it also includes anomalies in sensorial perception as diagnostic criteria given that individuals with ASD often display inexact sensorial processing, ranging from a lack of response to exaggerated reactions to sensorial stimuli [2,3]; thus, the clinical picture of ASD may vary in its symptomology and severity [4].

Emotional self-regulation (ER) is defined as the ability to control and modify one’s own emotional state, acting in consequence and facilitating adaptive behavior [5,6]. The process of controlling one’s own emotions can be automatic or intentional and may be a response to a current or antecedent situation [7]. ER strategies may be adaptive or maladaptive, the latter referring to psychopathologies and problematic behavior, and more prevalent among those with ASD [8].

Emotional regulation requires the recognition of one’s own emotional state, that is, the ability to identify and describe internal bodily sensations that signal a lack of physical homeostasis, spurring action to restore balance [6,9]. Similarly, the awareness of internal signals leads to an adequate understanding of one’s emotions, permitting the development of adequate emotional regulation strategies [6].

The research has shown that inadequate ER and maladaptive ER strategies are prevalent among those with ASD, both in children and adults [7,10,11]. Studies have found that children with ASD are less likely to employ adaptive coping strategies compared to children without ASD [12,13] and, in moments of frustration, show higher levels of negative affect, which persists despite calming efforts [12]. The research has also found that adolescents with ASD have a greater propensity for maladaptive ER strategies, such as avoidance, negation, rumination, and emotional numbness [14,15]. Similar results have been seen in adults with ASD, who more commonly use strategies such as suppression [16], resignation, and avoidance [12] compared to neurotypical adults [17].

This suggests that those with ASD lack the motivational component of ER, an essential aspect in emotional control. This is apparently reinforced by low emotional perception and self-control. Thus, those with ASD appear to lack the emotional self-perception necessary for effective ER [10]. Other aspects of impaired ER are intrinsic to ASD, such as difficulties in maintaining perspective and problem solving, lower response inhibition, deficits in emotional awareness, and the inadequate interpretation of social signals [7].

The research has identified a connection between alexithymia and ASD given that both types of clinical cases manifest similar difficulties. Individuals with alexithymia often have difficulties in recognizing emotions, establishing social relations, understanding the intentions of others, presenting limitations in moral decision making, etc. These characteristics, in conjunction with impaired communication and social skills, are commonly observed among those with ASD [18,19]. Alexithymia and ASD constitute a personality trait that makes them struggle when verbalizing internal states due to the lack of awareness and clarity that allows for understanding body sensations and translating them into feelings [18].

Studies have also shown that those with ASD and alexithymia derive less enjoyment in social interactions [20] and tend to feel and show less empathy [21]. However, alexithymia is not considered a psychiatric category in itself (that is, not included in the DSM-5) but rather a group of symptoms manifested in various other clinical diagnoses (eating disorders, addiction, mood disorders, ASD, etc.), which vary in severity depending on the individual [22,23]. The role of alexithymia with the broader symptomology of ASD is unclear [19], but the rates of alexithymia in the ASD population range from 40% to 65% [24]. Recently, Ben Hassen et al. explained that difficulties in ER, commonly associated with ASD, may in fact be due to the presence of alexithymia rather than ASD per se [25].

Because adequate emotional awareness is the foundation of ER [26], the socio-emotional deficiencies common among those with ASD may be related to difficulties in perceiving internal bodily signals [27]. There has been growing research into the notion of interoception, also referred to as the ‘eighth sense’. It is now clear that human beings are not limited to seven senses (sight, smell, taste, touch, hearing, proprioceptive, and vestibular senses) [6]. Interoception is associated with the awareness of physiological sensations, such as pain, body temperature, hunger, thirst, heart rate and breathing, muscular tension, etc., the internal condition of the body [28,29]. The interoceptive sense has receptors in various parts of the body, in the organs, skin, and muscles which capture sensorial information and send it to the brain, specifically the insular cortex or insula, where they are translated into a clear message of what the body is feeling and experiencing [28].

Interoception is a fundamental component of emotional experience. Poor interoception inhibits a person’s awareness of internal signals, and hence an inability to deploy effective ER strategies [6,28,30]. Moreover, recent findings suggest that alexithymia can best be characterized as a general failure of interoception rather than a specific affective impairment [31].

Empirical evidence of the relation between interoception and ASD is limited and inconsistent; however, impaired or deficient interception is common among those with ASD [6] who show much lower levels of interoceptive awareness than those without ASD [32,33], specifically in their awareness of internal physical sensations, such as thirst [34]. Persons with ASD show hyposensitivity, with difficulties in perceiving and identifying bodily sensations, such as pain, etc. [3,35].

Based on this theoretical framework, impaired interoception, the inability to identify and describe internal signals, can lead to alexithymia [19,31], found to be highly prevalent among those with ASD [36] and a contributing factor in maladaptive ER [7,37], aggravating the disorder and further complicating emotional self-expression and social relationships [31,32,38,39].

It was not until very recently that interoception scales were tested in the ASD population even though impaired interoception may be closely related to many of the deficits identified among this population, such as alexithymia and impaired emotional regulation, leading to difficulties in social relations. The aim of this study is to explore the relation between interoceptive capacity, emotional regulation, and alexithymia among people with ASD. Establishing evidence of this phenomenon is the first step toward developing interventions and effective treatments to enhance the quality of life of individuals with ASD and their families.

The present study has the following objectives: (1) to determine the characteristics of interoceptive confusion, alexithymia, and emotional regulation among a sample of participants with ASD, and to identify areas where this group scores significantly from the control group with neurotypical development (CG); (2) to analyze the association between the interoception scale with scales of alexithymia and the ER subscales (attention, clarity, and emotional repair) for each group (ASD and CG); and (3) to explore the relative impact of the variables alexithymia, ER subscales (attention, clarity, and emotional repair), and autism as a predictor of interoceptive confusion.

## 2. Materials and Methods

### 2.1. Participants

The initial sample was 160 participants, men and women; 70 of the participants were eliminated as they did not meet the inclusion criteria or dropped out during the test.

The final sample consisted of 68 men, of whom 48.53% (*n* = 33) were adults diagnosed with ASD between 21 and 58 years of age (M = 34.29; SD = 10.89) and 51.47% (*n* = 35) were adults with neurotypical development between the ages of 18 and 58 (M = 33.43; SD = 13.25). Although 22 women completed the questionnaires, they were not included in the analysis due to the acknowledged differences in clinical presentation between men and women [40], especially in relation to difficulties in emotional regulation and empathy.

The inclusion criteria for the ASD group were (a) being over the age of 18; (b) having a confirmed diagnosis of ASD; (c) having the necessary language ability to complete the self-evaluation questionnaire individually and with a score above 6 in the AQ-10 [41] (described below). The inclusion criteria for the CG were (a) being over the age of 18; (b) not being diagnosed with ASD or other clinical diagnosis; and (c) scoring below 6 in the screening version of the AQ-10 test.

The education levels of the participants were evaluated as an indirect measure of intelligence and verbal proficiency. For the CG, 5.7% (*n* = 2) had a primary education, 2.9% (*n* = 1) secondary education, 11.4% (*n* = 4) a baccalaureate, 8.6% (*n* = 3) had vocational training, and 71.4% (*n* = 25) had a university degree. For the ASD group, 3.0% (*n* = 1) had primary education, 12.1% (*n* = 4) secondary education, 18.2% (*n* = 6) a baccalaureate, 33.3% (*n* = 11) had vocational training, and 33.3% (*n* = 11) had a university degree. We assumed, therefore, that they were all equally capable of answering the survey.

### 2.2. Procedure

After receiving approval of the Ethics Committee of the Universidad Francisco de Vitoria (8/2021), the research was conducted through various entities in contact with persons with ASD and through social media. The questionnaires were completed online, and all participants provided their informed consent. Participation was voluntary and anonymous. Data were obtained using non-probability and snowball sampling methods.

### 2.3. Measures

#### 2.3.1. Sociodemographic Questionnaire

Collecting data about age, gender, level of education (no education, primary, secondary, baccalaureate, vocational training, university education), clinical diagnosis (yes, no, I don’t know, I don’t want to answer), age of diagnosis.

#### 2.3.2. Autism Spectrum Quotient (AQ-10)

The Autism Spectrum Quotient (AQ) is a 50-item questionnaire which evaluates symptoms of autism in 5 domains: social skills, communication skills, attention to detail, attention switching, and imagination [42]. In 2012, Allison et al. developed the AQ-10, a reduced version of 10 items [41]. The responses are scored on a 4-point Likert-type scale (1 = definitely disagree and 4 = definitely agree). Scores higher than 6 are considered an indication of possible ASD. The AQ-10 showed good internal consistency in the original study (α= 0.85). In the present sample, the AQ-10 showed adequate reliability (α = 0.69).

#### 2.3.3. Trait Meta-Mood Scale (TMMS)

This is a trait scale that evaluates meta-knowledge of emotional states, specifically the ability to perceive and regulate one’s own emotions [43]. It consists of 24 items on a 5-point Likert-type scale (1 = strongly disagree and 5 = strongly agree) that address three key dimensions of emotional intelligence with 8 items each: (1) emotional attention, (2) emotional clarity, and (3) emotional repair. According to the test, scores considered Limited for men in each subscale was as follows: emotional attention < 21, emotional clarity < 25, and emotional repair < 23. Scores were considered adequate when they range from 22 to 33 for emotional attention, 26 to 35 for emotional clarity, and 24 to 35 for emotional repair [43]. It was validated to Spanish population by Fernández-Berrocal et al. [44] The scale shows adequate internal consistency, α = 0.90, α = 0.90, and α = 0.86, respectively. For this research, the scale showed excellent internal consistency, both overall (α = 0.96) and in its respective subscales (α = 0.92, α = 0.95, and α = 0.92).

#### 2.3.4. Toronto Alexithymia Scale (TAS-20)

This is a self-reporting, 20-item instrument using a 5-point Likert-type scale (1 = strongly disagree and 5 = strongly agree) which evaluates alexithymia in 3 cognitive-affective areas: (1) difficulty in identifying feelings, (2) difficulty describing feelings, and (3) externally oriented thinking. Scores between 52 and 60 indicate possible alexithymia and a score of 61 indicates the presence of alexithymia [45]. The higher the score, the greater the inability to recognize one’s own emotions. It has a Spanish version by Martínez-Sánchez [46]. Internal consistency of the tool for this study was considered excellent (α = 0.82).

#### 2.3.5. Interoceptive Sensory Questionnaire (ISQ)

A self-reporting scale that measures interoceptive confusion among adults with ASD, using language that is natural and understandable for this population [38]. The tool consists of 20 items using a 7-point Likert-type scale (1 = totally untrue for me and 7 = total true for me). A score of 70 indicates interoceptive confusion, that is, greater difficulty in interception. Scores equal to or above 94 indicate a high degree of interoceptive confusion. The questionnaire was only available in its original language (English); a process of reverse translation was used to translate it into Spanish for the purposes of this study. The internal consistency of the instrument for the present study was excellent (α = 0.96), in line with data from [38].

### 2.4. Data Analysis

IBM’s SPSS Statistics software, version 25, was used for statistical analysis. Given the features of the sample, non-parametric method was used. Firstly, a descriptive analysis was made of the principal research variables, followed by a Mann–Whitney U test to analyze the differences between the groups for each scale. To analyze the relation between the variables, Spearman’s coefficient was used for each group. Finally, a multiple stepwise linear regression test was conducted to determine the predictive value of the variables for interoception.

## 3. Results

Firstly, the descriptive statistics were analyzed for the scales of interoceptive confusion, alexithymia, and ER for each group (ASD and CG). Table 1 shows that the scores of the ASD group were above the clinical cut-off point (≥ 61 for alexithymia; <21 for emotional attention; <25 for emotional clarity; and <23 for emotional repair) (see Table 1).

The analysis showed statistically significant differences between the groups (*p* < 0.01) in all dimensions of the study. The ASD group scored significantly higher in the scales of interoception (z = −5.18) and alexithymia (z = −4.18), while the CG scored higher in all the subscales for emotional regulation: emotional attention (z = −3.16), emotional clarity (z = −4.60), and emotional repair (z = −4.13) (see Table 1).

A non-parametric Spearman correlation test was used to analyze the association between the interoception scale with the scales of alexithymia and the subscales of ER for each group, which showed significant inverse correlations in the ASD group between interoceptive confusion and emotional clarity and a significant direct correlation with alexithymia (see Table 2). For the CG, a significant direct correlation was found between interoceptive confusion and alexithymia (see Table 3).

Finally, a multiple stepwise lineal regression was conducted to identify the principal predictors of interoception. A Durbin–Watson test was first performed to confirm the correlation of the variables, with a result of 1.71. The regression showed an R^2^ corrected equal to 0.61, indicating that the variables emotional clarity, alexithymia, and autism predict 61% of the variable interoception, with greater interceptive confusion corresponding to less emotional clarity (ER subscale) and with greater difficulties in alexithymia and autism corresponding to greater interoceptive confusion (See Table 4). The results of an ANOVA test confirm the goodness of fit of the model (*p* < 0.05).

## 4. Discussion

The purpose of this study was to evaluate interoceptive skills in relation to alexithymia among adult men with ASD, as these variables are associated with poor ER strategies [32,47,48].

Our study found that only the ASD group was suggestive of interoceptive confusion, in line with the presence of alexithymia, as well as limited emotional regulation. There were also statistically significant differences between the ASD and CG groups in the scales, confirming higher scores in the scales of alexithymia and interoceptive confusion and lower scores in the ER subscales for the ASD group. These results are in line with those of previous studies which found that people with ASD show greater levels of interoceptive confusion [34,49,50,51,52,53], greater symptoms of alexithymia [22,54], and worse ER strategies [7,16].

Regarding the association between the interoception scores, alexithymia, and the ER subscales, the study found a significant correlation in the ASD group between the interoceptive confusion and the alexithymia scale (positive relation) and the ER subscale of emotional clarity (inverse relation) in contrast with the results in the CG in which a significant correlation was found between the scale of interoception and alexithymia (positive correlation). Hence, these results are in coherence with those who found that alexithymia is closely associated with impaired ER because ER strategies necessarily require an initial awareness and identification of emotions [47,48]. This lack of internal emotional understanding also impairs the awareness of the feelings and emotions of others, hindering social interaction [18,55]. In fact, studies have found that persons with high scores in alexithymia often make use of ineffective ER strategies, such as suppression [48,54]. Along the same line, although scarce, some previous researchers found a relation between the presence of alexithymia and interoception both in the ASD population [31,56,57,58] and the non-clinical population [59]. Researchers also found that greater interoceptive confusion corresponds to poor ER strategies [6,16,60,61].

Scientific evidence showed that the interoceptive difficulties found among individuals with ASD are usually classified into three patterns [52]. Interoceptive hyposensitivity is where the person is incapable of noting the internal bodily signals or emotional state unless they are very intense; interoceptive hypersensitivity is where the person perceives internal sensations quickly and intensely, or feels various sensations at the same time, causing difficulties in determining which signals are the most important. However, there is a third pattern of interoceptive impairment that includes poor discrimination (sometimes accompanied by a lack of verbal accuracy), where the person is unable to identity the exact internal feeling because the experience is a vague or generalized sensation, with a limited ability to identity the meaning of this feeling [6,62]. This third phenomenon has led to confusing results in several studies [63,64]. As in the present study, sometimes these three patterns of interoception impediments make it hard to find a clear linear association within the data set, although the differences between the groups appeared at first.

One of our significant findings relates to the controversial relations between alexithymia, autism, ER, and interoception. Our study joins others [65] that know these variables tend to be correlated even though they are not yet considered diagnostic criteria and the relationship between them is not clear. One question underlying this study is which variables can be related to others so that future intervention can be more effective.

The other finding is that although these associations between these variables may be intuitive, the present study shows they did not occur in the same way in each group. Emotional regulation seems to be a more cognitive-based skill in the control group and a more physical-based skill in the ASD group.

Finally, it may be affirmed that emotional clarity (ER subscale), alexithymia, and autism are predictive variables of interoceptive confusion for 61% of the total sample. In contrast, the variables attention and emotional repair (ER subscales) are not predictors of interoception. These results suggest that greater levels of emotional clarity correspond to better interoception, while symptoms of alexithymia and autism correspond to greater interoceptive confusion. The present study adds evidence to the direct impact that interoceptive confusion may have on emotional dysfunction. In clinical practice, this may indicate a greater probability of interoceptive confusion among those showing signs of alexithymia and autism. The inverse relation between emotional clarity and interoception suggests that greater difficulties in emotional clarity (an inadequate understanding of one’s own emotions) might reflect higher levels of interoceptive confusion. Assessing interoception prior to intervention and integrating the results into the therapy may help diminish the alexithymia and emotional difficulties for those with ASD. Previous research has shown that training to develop interoception (improving interoceptive awareness) can enhance the capacity for effective emotional regulation [63,65,66].

These findings suggest that interoceptive awareness is the basis of more complex constructs, such as the recognition and expression of emotional states and ER. The perception and understanding of internal sensations is clearly a fundamental prerequisite for adequate emotional regulation [26,67,68]. Thus, interoceptive ability can be enhanced by the development of emotional clarity, reducing alexithymia among those diagnosed with ASD. This issue may have important implications for the prevention of social challenges that are extended to intimate relationships and identity when it comes to adolescence and adulthood [69,70]

On the whole, this study shows that a more precise description with a clinical approach should be considered when assessing and developing intervention programs for those with ASD to effectively address these difficulties, including training in interoceptive awareness, reducing alexithymia, and developing adaptive ER strategies [63].

### Limitations and Future Research

The present research is not without its limitations. The sample was relatively small, without the possibility of analyzing the variables of the study by gender. Future research should ideally use larger samples including women for more complete and generalizable results. Additionally, longitudinal studies of ASD populations should explore the possibility that the understanding of emotional and physical signals can vary throughout the life cycle. The present study offers data on adults only, and it is therefore unclear if these results can be extrapolated to children with ASD.

Another limitation was the use of self-reporting evaluation tools which produce biased or inaccurate results. Given that those with ASD often have difficulties in perceiving and identifying internal signals, analyzing problems, and expressing emotions [18,32,71], it may be instructive to use different evaluation instruments to compare scores based on mixed methods.

Additionally, this study was the first to use the Interoceptive Sensory Questionnaire with a Spanish sample. The good internal consistency (α = 0.95) indicates that this may be an effective tool for clinical assessment prior to intervention [52].

## 5. Conclusions

The study found that adult men with ASD have difficulty identifying internal bodily signals, with limitations in their ability to identify and regulate their emotional and physical state. We propose that future research explores the possibility of developing intervention programs based on increasing interoceptive awareness, focusing specifically on emotional clarity and alexithymia as these variables predict and influence interoceptive ability.

## Figures and Tables

**Table 1 behavsci-13-00312-t001:** Mann–Whitney U test for the ASD and CG groups.

Variable	ASD Group			GC Group			U	Z	*p*
	M	Me	SD	M	Me	SD			
Interoceptive confusion	68.90	67.00	27.96	35.69	33.00	12.01	155.50	−5.18	0.000
Alexithymia	63.15 *	62.00	9.30	50.80	50.00	12.15	237.00	−4.18	0.000
Emotional attention	20.67 *	20.00	9.03	27.17	28.00	7.27	320.50	−3.16	0.002
Emotional clarity	20.72 *	17.00	8.09	30.77	29.00	6.53	203.00	−4.60	0.000
Emotional repair	21.00 *	21.00	7.96	29.54	30.00	6.29	241.50	−4.13	0.000

M = mean; SD = standard deviation; U = Mann–Whitney U statistic; Z = Z statistic; *p* = level of significance; * = scores beyond clinical cut-off point.

**Table 2 behavsci-13-00312-t002:** Spearman’s correlation for the ASD group.

	Interoceptive Confusion	Emotional Attention	Emotional Clarity	Emotional Repair	Alexithymia
Interoceptive confusion	1	−0.195	−0.578 **	−0.286	0.502 **
Emotional attention		1	0.656 **	0.298	0.176
Emotional clarity			1	0.569 **	−0.165
Emotional repair				1	−0.121
Alexithymia					1

** The correlation is significant at level 0.01.

**Table 3 behavsci-13-00312-t003:** Spearman’s correlation for the CG.

	Interoceptive Confusion	Emotional Attention	Emotional Clarity	Emotional Repair	Alexithymia
Interoceptive confusion	1	0.008	−0.090	0.143	0.563 **
Emotional attention		1	−275	0.463 **	0.081
Emotional clarity			1	0.332	−0.146
Emotional repair				1	0.196
Alexithymia					1

** The correlation is significant at level 0.01.

**Table 4 behavsci-13-00312-t004:** Multiple stepwise linear regression coefficients for interoception.

Included Variable	R^2^	R^2^ Corrected	F (sig.)	*Β*	*t*	*p*
Step 3	0.782	0.611	33.55 (0.000)			
Emotional clarity				−0.328	−4.25	0.000
Alexithymia				0.375	4.35	0.000
Autism				0.253	2.68	0.009

R^2^ = determination coefficient; F (sig.) = F significance; β = regression coefficient; *t* = Student’s t; *p* = level of significance.

## Data Availability

The data presented in this study are available on request from the corresponding author.

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
