# Peer review of "Emotional Dysfunction and Interoceptive Challenges in Adults with Autism Spectrum Disorders"

_behavsci, 2023, doi:10.3390/bs13040312_

Round 1
Reviewer 1 Report
1. In the method section, the authors need to introduce the method of calculating the amount of participants to ensure that the included amount can achieve a certain statistical power.
2. In the procedure section, the authors did not report on specific ethical review bodies.
3. Is it possible to perform a more nuanced analysis of the subscale dimensions of these scales, as the available results are not very informative
4. Differences between ASD and control groups and correlations between variables seem to be expected, what updated knowledge the article provides? and the value and significance of the study may need to be more clearly emphasized (in both intro and discussion).
Author Response
Point 1: In the method section, the authors need to introduce the method of calculating the amount of participants to ensure that the included amount can achieve a certain statistical power.
Response 1: This was a descriptive, analytical study that was performed using a non probabilistic sample of convenience with a clinical sample ussually hard to access. Comparisons with control group enhance the analysis (since both groups are only different in presenting or not ASD) but we are aware that the conclusions have limitations. Therefore, this limitations are addressed at the end of the paper.
Point 2: In the procedure section, the authors did not report on specific ethical review bodies.
Response 2: Code and name of the committee has been included. (Line 187)
Point 3: Is it possible to perform a more nuanced analysis of the subscale dimensions of these scales, as the available results are not very informative
Response 3: In Table 1 the values that were found beyond the clinical cut-off point proposed by author’s scale were marked as *. The Discussion section now has more detailed interpretations about the subscales associations with all the other variables.
Point 4: Differences between ASD and control groups and correlations between variables seem to be expected, what updated knowledge the article provides? and the value and significance of the study may need to be more clearly emphasized (in both intro and discussion).
Response 4: One of the significace of our findings lays on the controversial relations between Alexitimia, Autism, Emotional Regulation and Interoception. Our study joins others (e.g. Mahler et al. , 2022) that knowing this variables tend to be correlated (although they are not yet considered diagnostic criteria) the relationship between them is not clear. One question underneath the study is which variables can be related to others, so that future intervention can be more effective.
The other finding is that although these association between these variables may be intuitive, our study showed they did not occur in the same way in each group. Emotional Regulation seems to be a more cognitive-based skill in the Control Group, and a more physical-based skill in the ASD group.
This might be not so clear in the paper, therefore it has been clarified.
Reviewer 2 Report
The topic sound quite interesting. The authors provided quite sufficient literature to frame the context. Some points should be clarified, in details:
1. The manuscript must be revised to eliminate linguistic typos (for example, line 84, line 102)
2. Line 42-43: this statement should be supported by a reference.
3. Please refer to other studies in an appropriate way (“et al.” should be used instead of “y cols.”).
4. In the Demographic questionnaire it is requested the level of education, but its usefulness is not investigated in the analysis. Which is the usefulness of such information?
5. Figure 1. It is not really a flow chart.
6. All works from other authors should be cited (lines 200-204).
7. Authors selected two different p-values for the different statistical tests they applied; is it necessary? It would have more scientific soundness to choose the same level of significance.
Author Response
Point 1: The manuscript must be revised to eliminate linguistic typos (for example, line 84, line 102)
Response 1: We revised the text and modified errors written in Spanish. The manuscript is correctly translated into English.
Point 2: Line 42-43: this statement should be supported by a reference.
Response 2: We include the reference by the end of the sentence (reference number [18-19]). We checked that all the sentences were well referenced.
Point 3: Please refer to other studies in an appropriate way (“et al.” should be used instead of “y cols.”).
Response 3: Changes were done following reviewer indications in line 76 (Ben Hassen et al. [25]) and line 100 (Elwin et al [35]).
Point 4: In the Demographic questionnaire it is requested the level of education, but its usefulness is not investigated in the analysis. Which is the usefulness of such information?
Response 4: We used the educational level as a control variable. As it was not possible to measure the IQ directly (due to anonymous participation), we collected information about the level of education instead.
Point 5: Figure 1. It is not really a flow chart.
Response 5: We eliminated the esqueme of participants recruitment because we included the description through the text in the Participants section.
Point 6: All works from other authors should be cited (lines 200-204).
Response 6: We checked that all the sentences were well referenced and modified the ones that were missing.
Point 7: Authors selected two different p-values for the different statistical tests they applied; is it necessary? It would have more scientific soundness to choose the same level of significance.
Response 7: In line 253 the reference to the p value was changed to 0.01 as it was thought to be always the same p-value. For the three tables, significant p values are always under 0.01.
Reviewer 3 Report
This paper describes a very interesting and very important research. It would gain more strength and be easier to read by improving the structure of the presentation.
The introduction is much too long and includes some information that would have been better to present to support the Discussion.
The Mat and Meths section must be improved by respecting the logical contain of each part of this section. The first part, with the description of the population enrolled in the study, includes data that should be in the Results section. Since there is a method for the inclusion and exclusion, it should be presented in the Mat and Meths section, but the final numbers of included and excluded people, should be presented in the Results section. As it is currently presented, some may expect a statistical analysis including an ITT and PP analysis, which would be a non-sense in such study. I recommend to stick to regular presentation methods and separate the method from the results (final structure of the enrolled population.
The authors should pay attention to the "comfort" of the readers and make it simple, when it can be. As an example, the writing of line 205 " mentioning that the scale has been "Translated to Spanish by (49)" would have been more easy going with mentioning the name of the authors of this translation, and then the number of the reference. It is just an example and a detail, but some other sentences of the same kind, may be simplified.
The Results section is too long, because it includes both comments (deserving to be in the Discussion section), but also methodology details, that should be mentioned in the Mat and Meths section. Some examples lines 248-252 should be in Discussion; 261-263 Meths and Stats; 272-279 are mixing Meth/Results and Discussion. The Results section must be as short and simple as possible, just presenting stratified results. This straification is also a good strategy to organize the Discussion, beginning with this stratification to emphasize the Discussion with the analysis of the global results, under the light of current science and clinical practice. It will, again, improve the comfort of the readers.
Author Response
Point 1: The introduction needs to be shorten by changing some information to support the Discussion
Response 1: Introduction has been shorten and some information has been changed to Discussion to support the results founded. We think the presentation of the ideas was improved.
Point 2: The Method part must be improved. The description of the population enrolled in the study includes data that should be in the results section. The final numbers of included and excluded people, should be presented in the Results section.
Response 2: We improved the Method section. Percetage of participants and level of education education were included as part of the Method section because it was been thought as evidence of homogeneous cognitive level and ability to understand the tests. Information that it was needed to control prior to compare the two groups in the relevant variables for the study (interoception, alexithymia and emotional regulation).
Point 3: The authors should simplify the writing to be easier to follow by the readers. Exmple: writing of line 205 instead of “the scale has been translated to Spanish by (49), would have been more easy going bith mentioning the name of the authors of this translation, and then the number of the reference. There are some other senteces like this
Response 3: Changes were done following reviewer indications in line 165, 173, 186.
Point 4: The Results section needs to be reorganize because it includes both comments (deserving to be in the Discussion section) but also methodology details, that should be mention in the Meths section. Some examples lines 248-252 should be in the Discussion. The stratified presentation of the results should be also follow in the Discussion.
Response 4: Changes were done following reviewer indications. Lines 248-252 were changed in order to remove the interpretation of the data in the Results section and it was added in the Discussion. We improved the coherence in the stratified presentation of the Results and Discussion.
Round 2
Reviewer 3 Report
Thank you for improving your paper. It is significantly easier to read and the results sound much more sensible. In my opinion, it would gain in quality, if you clearly differentiate Mat & Meths and Results. It is usual to consider the descritpion of the population enrolled in a study, as a result. In your paper, lines 126-131 and 140-146 (the beginning of the paragraph is line 139, that is clearly methodological for the whole sentence beginning on line 129 and finishing in the middle of line 140).